# Large Margin Deep Networks for Classification

**Gamaleldin F. Elsayed** [*]
Google Research

**Dilip Krishnan**
Google Research

**Hossein Mobahi**
Google Research

**Kevin Regan**
Google Research

**Samy Bengio**
Google Research
{gamaleldin, dilipkay, hmobahi, kevinregan, bengio}@google.com

## Abstract

We present a formulation of deep learning that aims at producing a large margin classifier. The notion of *margin*, minimum distance to a decision boundary, has served as the foundation of several theoretically profound and empirically successful results for both classification and regression tasks. However, most large margin algorithms are applicable only to shallow models with a preset feature representation; and conventional margin methods for neural networks only enforce margin at the output layer. Such methods are therefore not well suited for deep networks. In this work, we propose a novel loss function to impose a margin on any chosen set of layers of a deep network (including input and hidden layers). Our formulation allows choosing any $l_p$ norm ($p \geq 1$) on the metric measuring the margin. We demonstrate that the decision boundary obtained by our loss has nice properties compared to standard classification loss functions. Specifically, we show improved empirical results on the MNIST, CIFAR-10 and ImageNet datasets on multiple tasks: generalization from small training sets, corrupted labels, and robustness against adversarial perturbations. The resulting loss is general and complementary to existing data augmentation (such as random/adversarial input transform) and regularization techniques such as weight decay, dropout, and batch norm. [2]

## 1 Introduction

The large margin principle has played a key role in the course of machine learning history, producing remarkable theoretical and empirical results for classification (Vapnik, 1995) and regression problems (Drucker et al., 1997). However, exact large margin algorithms are only suitable for shallow models. In fact, for deep models, computation of the margin itself becomes intractable. This is in contrast to classic setups such as kernel SVMs, where the margin has an analytical form (the $l_2$ norm of the parameters). Desirable benefits of large margin classifiers include better generalization properties and robustness to input perturbations (Cortes & Vapnik, 1995; Bousquet & Elisseeff, 2002).

To overcome the limitations of classical margin approaches, we design a novel loss function based on a first-order approximation of the margin. This loss function is applicable to any network architecture (e.g., arbitrary depth, activation function, use of convolutions, residual networks), and complements existing general-purpose regularization techniques such as weight-decay, dropout and batch normalization.

---

[*]Work done as member of the Google AI Residency program https://ai.google/research/join-us/ai-residency/

[2]Code for the large margin loss function is released at https://github.com/google-research/google-research/tree/master/large_margin

We illustrate the basic idea of a large margin classifier within a toy setup in Figure 1. For demonstration purposes, consider a binary classification task and assume there is a model that can perfectly separate the data. Suppose the models is parameterized by vector $w$, and the model $g(x; w)$ maps the input vector $x$ to a real number, as shown in Figure 1(a); where the yellow region corresponds to positive values of $g(x; w)$ and the blue region to negative values; the red and blue dots represent training points from the two classes. Such $g$ is sometimes called a *discriminant function*. For a fixed $w$, $g(x; w)$ partitions the input space into two sets of regions, depending on whether $g(x; w)$ is positive or negative at those points. We refer to the boundary of these sets as the *decision boundary*, which can be characterized by $\{x \mid g(x; w) = 0\}$ when $g$ is a continuous function. For a fixed $w$, consider the distance of each training point to the decision boundary. We call the smallest non-negative such distance the *margin*. A large margin classifier seeks model parameters $w$ that attain the *largest* margin. Figure 1(b) shows the decision boundaries attained by our new loss (right), and another solution attained by the standard cross-entropy loss (left). The yellow squares show regions where the large margin solution better captures the correct data distribution.

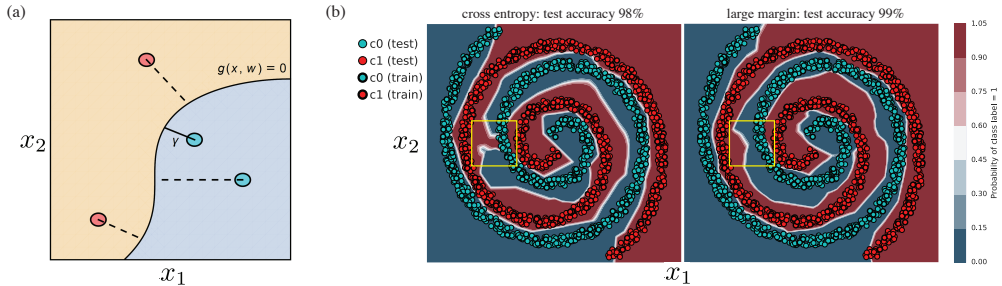

Figure 1: **Illustration of large margin.** (a) The distance of each training point to the decision boundary, with the shortest one being marked as $\gamma$. While the closest point to the decision boundary does not need to be unique, the value of shortest distance (i.e. $\gamma$ itself) is unique. (b) Toy example illustrating a good and a bad decision boundary obtained by optimizing a 4-layer deep network with cross-entropy loss (left), and with our proposed large margin loss (right). The two losses were trained for $10000$ steps on data shown in bold dots (train accuracy is $100\%$ for both losses). Accuracy on test data (light dots) is reported at the top of each figure. Note how the decision boundary is better shaped in the region outlined by the yellow squares. This figure is best seen in PDF.

Margin may be defined based on the values of $g$ (i.e. the output space) or on the input space. Despite similar naming, the two are very different. Margin based on output space values is the conventional definition. In fact, output margin can be computed exactly even for deep networks (Sun et al., 2015). In contrast, the margin in the input space is computationally intractable for deep models. Despite that, the input margin is often of more practical interest. For example, a large margin in the input space implies immunity to input perturbations. Specifically, if a classifier attains margin of $\gamma$, i.e. the decision boundary is at least $\gamma$ away from all the training points, then any perturbation of the input that is smaller than $\gamma$ will not be able to flip the predicted label. More formally, a model with a margin of $\gamma$ is robust to perturbations $x + \delta$ where $\text{sign}(g(x)) = \text{sign}(g(x + \delta))$, when $\|\delta\| < \gamma$. It has been shown that standard deep learning methods lack such robustness (Szegedy et al., 2013).

In this work, our main contribution is to derive a new loss for obtaining a large margin classifier for deep networks, where the margin can be based on any $l_p$-norm ($p \geq 1$), and the margin may be defined on any chosen set of layers of a network. We empirically evaluate our loss function on deep networks across different applications, datasets and model architectures. Specifically, we study the performance of these models on tasks of adversarial learning, generalization from limited training data, and learning from data with noisy labels. We show that the proposed loss function consistently outperforms baseline models trained with conventional losses, e.g. for adversarial perturbation, we outperform common baselines by up to $21\%$ on MNIST, $14\%$ on CIFAR-10 and $11\%$ on Imagenet.

## 2   Related Work

Prior work (Liu et al., 2016; Sun et al., 2015; Sokolic et al., 2016; Liang et al., 2017) has explored the benefits of encouraging large margin in the context of deep networks. Sun et al. (2015) state that

cross-entropy loss does not have margin-maximization properties, and add terms to the cross-entropy loss to encourage large margin solutions. However, these terms encourage margins only at the output layer of a deep neural network. Other recent work (Soudry et al., 2017), proved that one can attain max-margin solution by using cross-entropy loss with stochastic gradient descent (SGD) optimization. Yet this was only demonstrated for linear architecture, making it less useful for deep, nonlinear networks. Sokolic et al. (2016) introduced a regularizer based on the Jacobian of the loss function with respect to network layers, and demonstrated that their regularizer can lead to larger margin solutions. This formulation only offers $L_2$ distance metrics and therefore may not be robust to deviation of data based on other metrics (e.g., adversarial perturbations). In contrast, our work formulates a loss function that directly maximizes the margin at any layer, including input, hidden and output layers. Our formulation is general to margin definitions in different distance metrics (e.g. $l_1$, $l_2$, and $l_\infty$ norms). We provide empirical evidence of superior performance in generalization tasks with limited data and noisy labels, as well as robustness to adversarial perturbations. Finally, Hein & Andriushchenko (2017) propose a linearization similar to ours, but use a very different loss function for optimization. Their setup and optimization are specific to the adversarial robustness scenario, whereas we also consider generalization and noisy labels; their resulting loss function is computationally expensive and possibly difficult to scale to large problems such as Imagenet. Matyasko & Chau (2017) also derive a similar linearization and apply it to adversarial robustness with promising results on MNIST and CIFAR-10.

In real applications, training data is often not as copious as we would like, and collected data might have noisy labels. Generalization has been extensively studied as part of the semi-supervised and few-shot learning literature, e.g. (Vinyals et al., 2016; Rasmus et al., 2015). Specific techniques to handle noisy labels for deep networks have also been developed (Sukhbaatar et al., 2014; Reed et al., 2014). Our margin loss provides generalization benefits and robustness to noisy labels and is complementary to these works. Deep networks are susceptible to adversarial attacks (Szegedy et al., 2013) and a number of attacks (Papernot et al., 2017; Sharif et al., 2016; Hosseini et al., 2017), and defenses (Kurakin et al., 2016; Madry et al., 2017; Guo et al., 2017; Athalye & Sutskever, 2017) have been developed. A natural benefit of large margins is robustness to adversarial attacks, as we show empirically in Sec. 4.

## 3   Large Margin Deep Networks

Consider a classification problem with $n$ classes. Suppose we use a function $f_i : \mathcal{X} \to \mathbb{R}$, for $i = 1, \ldots, n$ that generates a prediction score for classifying the input vector $\boldsymbol{x} \in \mathcal{X}$ to class $i$. The predicted label is decided by the class with maximal score, i.e. $i^* = \arg\max_i f_i(\boldsymbol{x})$.

Define the *decision boundary* for each class pair $\{i, j\}$ as:

$$\mathcal{D}_{\{i,j\}} \triangleq \{\boldsymbol{x} \mid f_i(\boldsymbol{x}) = f_j(\boldsymbol{x})\} \tag{1}$$

Under this definition, the distance of a point $\boldsymbol{x}$ to the decision boundary $\mathcal{D}_{\{i,j\}}$ is defined as the smallest displacement of the point that results in a score tie:

$$d_{f,\boldsymbol{x},\{i,j\}} \triangleq \min_{\boldsymbol{\delta}} \|\boldsymbol{\delta}\|_p \quad \text{s.t.} \quad f_i(\boldsymbol{x} + \boldsymbol{\delta}) = f_j(\boldsymbol{x} + \boldsymbol{\delta}) \tag{2}$$

Here $\|.\|_p$ is any $l_p$ norm ($p \geq 1$). Using this distance, we can develop a large margin loss. We start with a training set consisting of pairs $(\boldsymbol{x}_k, y_k)$, where the label $y_k \in \{1, \ldots, n\}$. We penalize the *displacement* of each $\boldsymbol{x}_k$ to satisfy the margin constraint for separating class $y_k$ from class $i$ ($i \neq y_k$). This implies using the following loss function:

$$\max\{0, \gamma + d_{f,\boldsymbol{x}_k,\{i,y_k\}} \operatorname{sign}\left(f_i(\boldsymbol{x}_k) - f_{y_k}(\boldsymbol{x}_k)\right)\}, \tag{3}$$

where the $\operatorname{sign}(.)$ adjusts the polarity of the distance. The intuition is that, if $\boldsymbol{x}_k$ is already correctly classified, then we only want to ensure it has distance $\gamma$ from the decision boundary, and penalize proportional to the distance $d$ it falls short (so the penalty is $\max\{0, \gamma - d\}$). However, if it is misclassified, we also want to penalize the point for not being correctly classified. Hence, the penalty includes the distance $\boldsymbol{x}_k$ needs to travel to reach the decision boundary as well as another $\gamma$ distance to travel on the correct side of decision boundary to attain $\gamma$ margin. Therefore, the penalty becomes $\max\{0, \gamma + d\}$. In a multiclass setting, we *aggregate* individual losses arising from each $i \neq y_k$ by some aggregation operator $\mathscr{A}$:

$$\mathscr{A}_{i \neq y_k} \max\{0, \gamma + d_{f,\boldsymbol{x}_k,\{i,y_k\}} \operatorname{sign}\left(f_i(\boldsymbol{x}_k) - f_{y_k}(\boldsymbol{x}_k)\right)\} \tag{4}$$

In this paper we use two aggregation operators, namely the max operator $\max$ and the sum operator $\sum$. In order to learn $f_i$'s, we assume they are parameterized by a vector $\boldsymbol{w}$ and should use the notation $f_i(\boldsymbol{x}; \boldsymbol{w})$; for brevity we keep using the notation $f_i(\boldsymbol{x})$. The goal is to minimize the loss w.r.t. $\boldsymbol{w}$:

$$\boldsymbol{w}^* \triangleq \arg\min_{\boldsymbol{w}} \sum_k \mathscr{A}_{i \neq y_k} \max\{0, \gamma + d_{f, \boldsymbol{x}_k, \{i, y_k\}} \, \text{sign}\,(f_i(\boldsymbol{x}_k) - f_{y_k}(\boldsymbol{x}_k))\} \tag{5}$$

The above formulation depends on $d$, whose exact computation from (2) is intractable when $f_i$'s are nonlinear. Instead, we present an approximation to $d$ by *linearizing* $f_i$ w.r.t. $\boldsymbol{\delta}$ around $\boldsymbol{\delta} = \boldsymbol{0}$.

$$\tilde{d}_{f, \boldsymbol{x}, \{i, j\}} \triangleq \min_{\boldsymbol{\delta}} \|\boldsymbol{\delta}\|_p \quad \text{s.t.} \quad f_i(\boldsymbol{x}) + \langle \boldsymbol{\delta}, \nabla_{\boldsymbol{x}} f_i(\boldsymbol{x}) \rangle = f_j(\boldsymbol{x}) + \langle \boldsymbol{\delta}, \nabla_{\boldsymbol{x}} f_j(\boldsymbol{x}) \rangle \tag{6}$$

This problem now has the following *closed form* solution (see supplementary for proof):

$$\tilde{d}_{f, \boldsymbol{x}, \{i, j\}} = \frac{|f_i(\boldsymbol{x}) - f_j(\boldsymbol{x})|}{\|\nabla_{\boldsymbol{x}} f_i(\boldsymbol{x}) - \nabla_{\boldsymbol{x}} f_j(\boldsymbol{x})\|_q} \, , \tag{7}$$

where $\|.\|_q$ is the *dual-norm* of $\|.\|_p$. $l_q$ is the dual norm of $l_p$ when it satisfies $q \triangleq \frac{p}{p-1}$ (Boyd & Vandenberghe, 2004). For example if distances are measured w.r.t. $l_1$, $l_2$, or $l_\infty$ norm, the norm in (7) will respectively be $l_\infty$, $l_2$, or $l_1$ norm. Using the linear approximation, the loss function becomes:

$$\hat{\boldsymbol{w}} \triangleq \arg\min_{\boldsymbol{w}} \sum_k \mathscr{A}_{i \neq y_k} \max\{0, \gamma + \frac{|f_i(\boldsymbol{x}_k) - f_{y_k}(\boldsymbol{x}_k)|}{\|\nabla_{\boldsymbol{x}} f_i(\boldsymbol{x}_k) - \nabla_{\boldsymbol{x}} f_{y_k}(\boldsymbol{x}_k)\|_q} \, \text{sign}\,(f_i(\boldsymbol{x}_k) - f_{y_k}(\boldsymbol{x}_k))\} \tag{8}$$

This further simplifies to the following problem:

$$\hat{\boldsymbol{w}} \triangleq \arg\min_{\boldsymbol{w}} \sum_k \mathscr{A}_{i \neq y_k} \max\{0, \gamma + \frac{f_i(\boldsymbol{x}_k) - f_{y_k}(\boldsymbol{x}_k)}{\|\nabla_{\boldsymbol{x}} f_i(\boldsymbol{x}_k) - \nabla_{\boldsymbol{x}} f_{y_k}(\boldsymbol{x}_k)\|_q}\} \tag{9}$$

In (Huang et al., 2015), (7) has been derived (independently of us) to facilitate adversarial training with different norms. In contrast, we develop a novel margin-based loss function that uses this distance metric at multiple hidden layers, and show benefits for a wide range of problems. In the supplementary material, we show that (7) coincides with an SVM for the special case of a linear classifier.

### 3.1 Margin for Hidden Layers

The classic notion of margin is defined based on the distance of *input* samples from the decision boundary; in shallow models such as SVM, input/output association is the only way to define a margin. In deep networks, however, the output is shaped from input by going through a number of transformations (layers). In fact, the activations at each intermediate layer could be interpreted as some intermediate representation of the data for the following part of the network. Thus, we can define the margin based on any intermediate representation and the ultimate decision boundary. We leverage this structure to enforce that the entire representation maintain a large margin with the decision boundary. The idea then, is to simply replace the input $\boldsymbol{x}$ in the margin formulation (9) with the intermediate representation of $\boldsymbol{x}$. More precisely, let $\boldsymbol{h}_\ell$ denote the output of the $\ell$'th layer ($\boldsymbol{h}_0 = \boldsymbol{x}$) and $\gamma_\ell$ be the margin enforced for its corresponding representation. Then the margin loss (9) can be adapted as below to incorporate intermediate margins (where the $\epsilon$ in the denominator is used to prevent numerical problems, and is set to a small value such as $10^{-6}$ in practice):

$$\hat{\boldsymbol{w}} \triangleq \arg\min_{\boldsymbol{w}} \sum_{\ell, k} \mathscr{A}_{i \neq y_k} \max\{0, \gamma_\ell + \frac{f_i(\boldsymbol{x}_k) - f_{y_k}(\boldsymbol{x}_k)}{\epsilon + \|\nabla_{\boldsymbol{h}_\ell} f_i(\boldsymbol{x}_k) - \nabla_{\boldsymbol{h}_\ell} f_{y_k}(\boldsymbol{x}_k)\|_q}\} \tag{10}$$

## 4 Experiments

Here we provide empirical results using formulation (10) on a number of tasks and datasets. We consider the following datasets and models: a deep convolutional network for MNIST (LeCun et al., 1998), a deep residual convolutional network for CIFAR-10 (Zagoruyko & Komodakis, 2016) and an Imagenet model with the Inception v3 architecture (Szegedy et al., 2016). Details of the architectures and hyperparameter settings are provided in the supplementary material. Our code was written in Tensorflow (Abadi et al., 2016). The tasks we consider are: training with noisy labels, training with limited data, and defense against adversarial perturbations. In all these cases, we expect that the presence of a large margin provides robustness and improves test accuracies. As shown below, this is indeed the case across all datasets and scenarios considered.

## 4.1 Optimization of Parameters

Our loss function (10) differs from the cross-entropy loss due to the presence of gradients *in the loss itself*. We compute these gradients for each class $i \neq y_k$ ($y_k$ is the true label corresponding to sample $\boldsymbol{x}_k$). To reduce computational cost, we choose a subset of the total number of classes. We pick these classes by choosing $i \neq y_k$ that have the highest value from the forward propagation step. For MNIST and CIFAR-10, we used all 9 (other) classes. For Imagenet we used only 1 class $i \neq y_k$ (increasing $k$ increased computational cost without helping performance). The backpropagation step for parameter updates requires the computation of second-order mixed gradients. To further reduce computation cost to a manageable level, we use a first-order Taylor approximation to the gradient with respect to the weights. This approximation simply corresponds to treating the denominator $(\|\nabla_{\boldsymbol{h}_\ell} f_i(\boldsymbol{x}_k) - \nabla_{\boldsymbol{h}_\ell} f_{y_k}(\boldsymbol{x}_k)\|_q)$ in (10) as a constant with respect to $\boldsymbol{w}$ for backpropagation. The value of $\|\nabla_{\boldsymbol{h}_\ell} f_i(\boldsymbol{x}_k) - \nabla_{\boldsymbol{h}_\ell} f_{y_k}(\boldsymbol{x}_k)\|_q$ is recomputed at every forward propagation step. We compared performance with and without this approximation for MNIST and found minimal difference in accuracy, but significantly higher GPU memory requirement due to the computation of second-order mixed derivatives without the approximation (a derivative with respect to activations, followed by another with respect to weights). Using these optimizations, we found, for example, that training is around $20\%$ to $60\%$ more expensive in wall-clock time for the margin model compared to cross-entropy, measured on the same NVIDIA p100 GPU (but note that there is no additional cost at inference time). Finally, to improve stability when the denominator is small, we found it beneficial to clip the loss at some threshold. We use standard optimizers such as RMSProp (Tieleman & Hinton, 2012).

## 4.2 MNIST

We train a 4 hidden-layer model with 2 convolutional layers and 2 fully connected layers, with rectified linear unit (ReLu) activation functions, and a softmax output layer. The first baseline model uses a cross-entropy loss function, trained with stochastic gradient descent optimization with momentum and learning rate decay. A natural question is whether having a large margin loss defined at the network output such as the standard hinge loss could be sufficient to give good performance. Therefore, we trained a second baseline model using a hinge loss combined with a small weight of cross-entropy.

The large margin model has the same architecture as the baseline, but we use our new loss function in formulation (10). We considered margin models using an $l_\infty$, $l_1$ or $l_2$ norm on the distances, respectively. For each norm, we train a model with margin either only on the input layer, or on all hidden layers and the output layer. Thus there are 6 margin models in all. For models with margin at all layers, the hyperparameter $\gamma_l$ is set to the same value for each layer (to reduce the number of hyperparameters). Furthermore, we observe that using a weighted sum of margin and cross-entropy facilitates training and speeds up convergence [3]. We tested all models with both stochastic gradient descent with momentum, and RMSProp (Hinton et al.) optimizers and chose the one that worked best on the validation set. In case of cross-entropy and hinge loss we used momentum, in case of margin models for MNIST, we used RMSProp with no momentum.

For all our models, we perform a hyperparameter search including with and without dropout, with and without weight decay and different values of $\gamma_l$ for the margin model (same value for all layers where margin is applied). We hold out $5,000$ samples of the training set as a validation set, and the remaining $55,000$ samples are used for training. The full evaluation set of $10,000$ samples is used for reporting all accuracies. Under this protocol, the cross-entropy and margin models trained on the $55,000$ sample training set achieves a *test* accuracy of $99.4\%$ and the hinge loss model achieve $99.2\%$.

### 4.2.1 Noisy Labels

In this experiment, we choose, for each training sample, whether to flip its label with some other label chosen at random. E.g. an instance of "1" may be labeled as digit "6". The percentage of such flipped labels varies from $0\%$ to $80\%$ in increments of $20\%$. Once a label is flipped, that label is fixed throughout training. Fig. 2(left) shows the performance of the best performing 4 (all layer margin

and cross-entropy) of the 8 algorithms, with test accuracy plotted against noise level. It is seen that the margin $l_1$ and $l_2$ models perform better than cross-entropy across the entire range of noise levels, while the margin $l_\infty$ model is slightly worse than cross-entropy. In particular, the margin $l_2$ model achieves a evaluation accuracy of $96.4\%$ at $80\%$ label noise, compared to $93.9\%$ for cross-entropy. The input only margin models were outperformed by the all layer margin models and are not shown in Fig. 2. We find that this holds true across all our tests. The performance of all 8 methods is shown in the supplementary material.

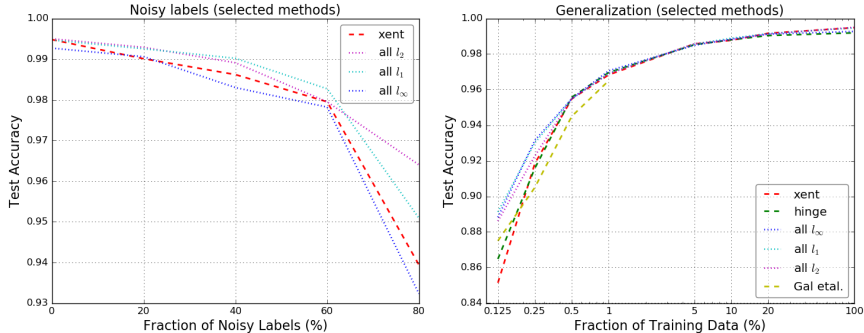

Figure 2: Performance of MNIST models on: (left) noisy label tasks and (right) generalization tasks.

### 4.2.2 Generalization

In this experiment we consider models trained with significantly lower amounts of training data. This is a problem of practical importance, and we expect that a large margin model should have better generalization abilities. Specifically, we randomly remove some fraction of the training set, going down from $100\%$ of training samples to only $0.125\%$, which is 68 samples. In Fig. 2(right), the performance of cross-entropy, hinge and margin (all layers) is shown. The test accuracy is plotted against the fraction of data used for training. We also show the generalization results of a Bayesian active learning approach presented in (Gal et al., 2017). The all-layer margin models outperform both cross-entropy and (Gal et al., 2017) over the entire range of testing, and the amount by which the margin models outperform increases as the dataset size decreases. The all-layer $l_\infty$-margin model outperforms cross-entropy by around $3.7\%$ in the smallest training set of 68 samples. We use the same randomly drawn training set for all models.

### 4.2.3 Adversarial Perturbation

Beginning with (Goodfellow et al., 2014), a number of papers (Papernot et al., 2016; Kurakin et al., 2016; Moosavi-Dezfooli et al., 2016) have examined the presence of adversarial examples that can "fool" deep networks. These papers show that there exist small perturbations to images that can cause a deep network to misclassify the resulting perturbed examples. We use the Fast Gradient Sign Method (FGSM) and the iterative version (IFGSM) of perturbation introduced in (Goodfellow et al., 2014; Kurakin et al., 2016) to generate adversarial examples[4]. Details of FGSM and IFGSM are given in the supplementary.

For each method, we generate a set of perturbed adversarial examples using one network, and then measure the accuracy of the same (white-box) or another (black-box) network on these examples. Fig. 3 (left, middle) shows the performance of the 8 models for IFGSM attacks (which are stronger than FGSM). FGSM performance is given in the supplementary. We plot test accuracy against different values of $\epsilon$ used to generate the adversarial examples. In both FGSM and IFGSM scenarios, all margin models significantly outperform cross-entropy, with the all-layer margin models outperform the input-only margin, showing the benefit of margin at hidden layers. This is not surprising as the adversarial attacks are specifically defined in input space. Furthermore, since FGSM/IFGSM are defined in the $l_\infty$ norm, we see that the $l_\infty$ margin model performs the best among the three norms. In the supplementary, we also show the white-box performance of the method from (Madry et al., 2017) [5], which is an algorithm specifically designed for adversarial defenses against FGSM attacks. One of the margin models outperforms this method, and another is very competitive. For black box,

the attacker is a cross-entropy model. It is seen that the margin models are robust against black-box attacks, significantly outperforming cross-entropy. For example at $\epsilon = 0.1$, cross-entropy is at $67\%$ accuracy, while the best margin model is at $90\%$.

Kurakin et al. (2016) suggested adversarial training as a defense against adversarial attacks. This approach augments training data with adversarial examples. However, they showed that adding FGSM examples in this manner, often do not confer robustness to IFGSM attacks, and is also computationally costly. Our margin models provide a mechanism for robustness that is independent of the type of attack. Further, our method is complementary and can still be used with adversarial training. To demonstrate this, Fig. 3 (right) shows the improved performance of the $l_\infty$ model compared to the cross-entropy model for black-box attacks from a cross-entropy model, when the models are adversarially trained. While the gap between cross-entropy and margin models is reduced in this scenario, we continue to see greater performance from the margin model at higher values of $\epsilon$. Importantly, we saw no benefit for the generalization or noisy label tasks from adversarial training - thus showing that this type of data augmentation provides very specific robustness. In the supplementary, we also show the performance against input corrupted with varying levels of Gaussian noise, showing the benefit of margin for this type of perturbation as well.

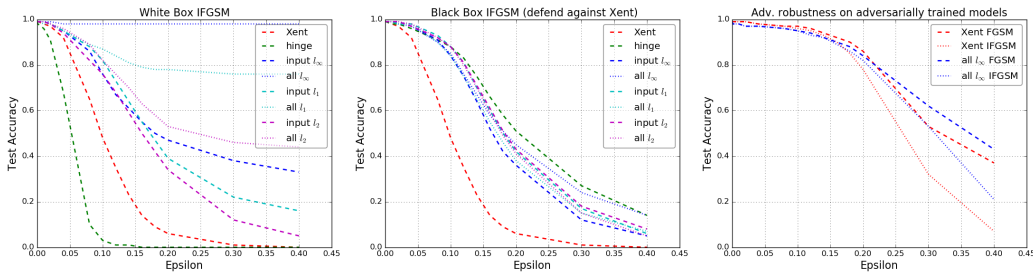

Figure 3: Robustness of MNIST models to adversarial attacks: (Left) White-box IFGSM; (Middle) Black-box IFGSM; (Right) Black-box performance of adversarially trained models.

## 4.3 CIFAR-10

Next, we test our models for the same tasks on CIFAR-10 dataset (Krizhevsky & Hinton, 2009). We use the ResNet model proposed in Zagoruyko & Komodakis (2016), consisting of an input convolutional layer, 3 blocks of residual convolutional layers where each block containing 9 layers, for a total of 58 convolutional layers. Similar to MNIST, we set aside $10\%$ of the training data for validation, leaving a total of $45,000$ training samples, and $5,000$ validation samples. We train margin models with multiple layers of margin, choosing 5 evenly spaced layers (input layer, output layer and 3 other convolutional layers in the middle) across the network. We perform a hyper-parameter search across margin values. We also train with data augmentation (random image transformations such as cropping and contrast/hue changes). Hyperparameter details are provided in the supplementary material. With these settings, we achieve a baseline accuracy of around $90\%$ for the following 5 models: cross-entropy, hinge, margin $l_\infty$, $l_1$ and $l_2$[6]

### 4.3.1 Noisy Labels

Fig. 4 (left) shows the performance of the 5 models under the same noisy label regime, with fractions of noise ranging from $0\%$ to $80\%$. The margin $l_\infty$ and $l_2$ models consistently outperforms cross-entropy by $4\%$ to $10\%$ across the range of noise levels.

### 4.3.2 Generalization

Fig. 4(right) shows the performance of the 5 CIFAR-10 models on the generalization task. We consistently see superior performance of the $l_1$ and $l_\infty$ margin models w.r.t. cross-entropy, especially as the amount of data is reduced. For example at $5\%$ and $1\%$ of the total data, the $l_1$ margin model outperforms the cross-entropy model by $2.5\%$.

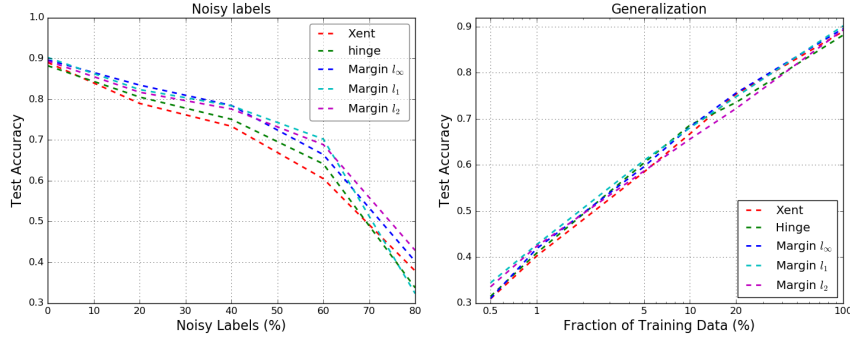

Figure 4: Performance of CIFAR-10 models on noisy data (left) and limited data (right).

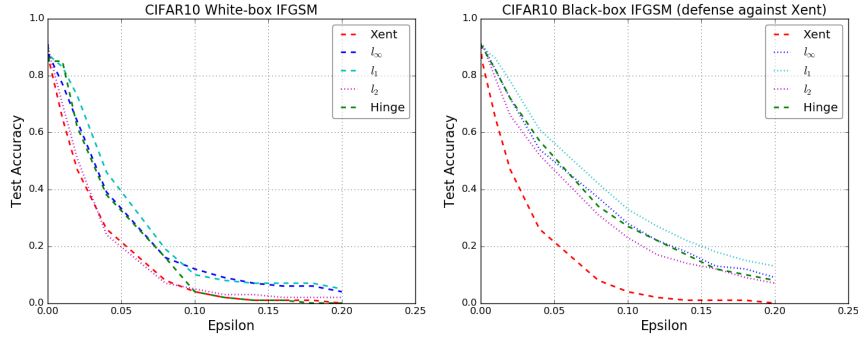

Figure 5: Performance of CIFAR-10 models on IFGSM adversarial examples.

### 4.3.3 Adversarial Perturbations

Fig. 5 shows the performance of cross-entropy and margin models for IFGSM attacks, for both white-box and black box scenarios. The $l_1$ and $l_\infty$ margin models perform well for both sets of attacks, giving a clear boost over cross-entropy. For $\epsilon = 0.1$, the $l_1$ model achieves an improvement over cross-entropy of about $14\%$ when defending against a cross-entropy attack. Another approach for robustness is in (Cisse et al., 2017), where the Lipschitz constant of network layers is kept small, thereby directly insulating the network from small perturbations. Our models trained with margin significantly outperform their reported results in Table 1 for CIFAR-10. For an SNR of 33 (as computed in their paper), we achieve 82% accuracy compared to 69.1% by them (for non-adversarial training), a 18.7 % relative improvement.

### 4.4 Imagenet

We tested our $l_1$ margin model against cross-entropy for a full-scale Imagenet model based on the Inception architecture (Szegedy et al., 2016), with data augmentation. Our margin model and cross-entropy achieved a top-1 validation precision of $78\%$ respectively, close to the $78.8\%$ reported in (Szegedy et al., 2016). We test the Imagenet models for white-box FGSM and IFGSM attacks, as well as for black-box attacks defending against cross-entropy model attacks. Results are shown in Fig. 6. We see that the margin model consistently outperforms cross-entropy for black and white box FGSM and IFGSM attacks. For example at $\epsilon = 0.1$, we see that cross-entropy achieves a white-box FGSM accuracy of 33%, whereas margin achieves $44\%$ white-box accuracy and $59\%$ black-box accuracy. Note that our FGSM accuracy numbers on the cross-entropy model are quite close to that achieved in (Kurakin et al., 2016) (Table 2, top row); also note that we use a wider range of $\epsilon$ in our experiments.

## 5 Discussion

We have presented a new loss function inspired by the theory of large margin that is amenable to deep network training. This new loss is flexible and can establish a large margin that can be defined on input, hidden or output layers, and using $l_\infty$, $l_1$, and $l_2$ distance definitions. Models trained with

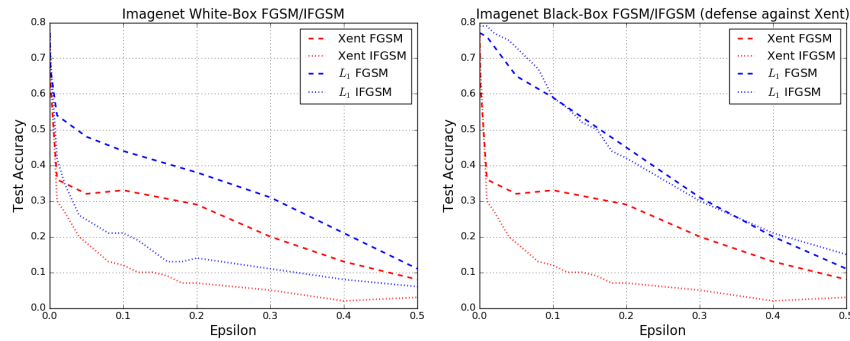

Figure 6: Imagenet white-box/black-box performance on adversarial examples.

this loss perform well in a number of practical scenarios compared to baselines on standard datasets. The formulation is independent of network architecture and input domain and is complementary to other regularization techniques such as weight decay and dropout. Our method is computationally practical: for Imagenet, our training was about 1.6 times more expensive than cross-entropy (per step). Finally, our empirical results show the benefit of margin at the hidden layers of a network.

## Footnotes

[3]We emphasize that the performance achieved with this combination cannot be obtained by cross-entropy alone, as shown in the performance plots.

[4]There are also other methods of generating adversarial perturbations, not considered here.

[5]We used $\epsilon$ values provided by the authors.

[6]With a better CIFAR network WRN-40-10 from Zagoruyko & Komodakis (2016), we were able to achieve $95\%$ accuracy on full data.

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
