[Supplementary Material]

# Large Margin Deep Networks for Classification: Supplementary Material

**Gamaleldin F. Elsayed**
Google Research

**Dilip Krishnan**
Google Research

**Hossein Mobahi**
Google Research

**Kevin Regan**
Google Research

**Samy Bengio**
Google Research

{gamaleldin, dilipkay, hmobahi, kevinregan, bengio}@google.com

## 1 Derivation of Equation (7) in paper

Consider the following optimization problem,

$$d \triangleq \min_{\boldsymbol{\delta}} \|\boldsymbol{\delta}\|_p \quad \text{s.t.} \quad \boldsymbol{a}^T \boldsymbol{\delta} = b \tag{1}$$

We show that $d$ has the form $d = \frac{|b|}{\|\boldsymbol{a}\|_*}$, where $\|.\|_*$ is the ***dual-norm*** of $\|.\|_p$. For a given norm $\|.\|_p$, the length of any vector $\boldsymbol{u}$ w.r.t. to the dual norm is defined as $\|\boldsymbol{u}\|_* \triangleq \max_{\|\boldsymbol{v}\|_p \leq 1} \boldsymbol{u}^T \boldsymbol{v}$. Since $\boldsymbol{u}^T \boldsymbol{v}$ is linear, the maximizer is attained at the boundary of the feasible set, i.e. $\|\boldsymbol{v}\|_p = 1$. Therefore, it follows that:

$$\|\boldsymbol{u}\|_* = \max_{\|\boldsymbol{v}\|_p = 1} \boldsymbol{u}^T \boldsymbol{v} = \max_{\boldsymbol{v}} \boldsymbol{u}^T \frac{\boldsymbol{v}}{\|\boldsymbol{v}\|_p} = \max_{\boldsymbol{v}} |\boldsymbol{u}^T \frac{\boldsymbol{v}}{\|\boldsymbol{v}\|_p}|, \tag{2}$$

In particular for $\boldsymbol{u} = \boldsymbol{a}$ and $\boldsymbol{v} = \boldsymbol{\delta}$:

$$\|\boldsymbol{a}\|_* = \max_{\boldsymbol{\delta}} |\boldsymbol{a}^T \frac{\boldsymbol{\delta}}{\|\boldsymbol{\delta}\|_p}| \tag{3}$$

So far we have just used properties of the dual norm. In order to prove our result, we start from the trivial identity (assuming $\|\boldsymbol{\delta}\|_p \neq 0$ which is guaranteed when $b \neq 0$): $\boldsymbol{a}^T \boldsymbol{\delta} = \|\boldsymbol{\delta}\|_p (\frac{\boldsymbol{a}^T \boldsymbol{\delta}}{\|\boldsymbol{\delta}\|_p})$. Consequently, $|\boldsymbol{a}^T \boldsymbol{\delta}| = \|\boldsymbol{\delta}\|_p |\frac{\boldsymbol{a}^T \boldsymbol{\delta}}{\|\boldsymbol{\delta}\|_p}|$. Applying the constraint $\boldsymbol{a}^T \boldsymbol{\delta} = b$, we obtain $|b| = \|\boldsymbol{\delta}\|_p |\frac{\boldsymbol{a}^T \boldsymbol{\delta}}{\|\boldsymbol{\delta}\|_p}|$. Thus, we can write,

$$\|\boldsymbol{\delta}\|_p = \frac{|b|}{|\frac{\boldsymbol{a}^T \boldsymbol{\delta}}{\|\boldsymbol{\delta}\|_p}|} \implies \min_{\boldsymbol{\delta}} \|\boldsymbol{\delta}\|_p = \min_{\boldsymbol{\delta}} \frac{|b|}{|\frac{\boldsymbol{a}^T \boldsymbol{\delta}}{\|\boldsymbol{\delta}\|_p}|} . \tag{4}$$

We thus continue as below (using (3) in the last step):

$$\min_{\boldsymbol{\delta}} \|\boldsymbol{\delta}\|_p = \min_{\boldsymbol{\delta}} \frac{|b|}{|\frac{\boldsymbol{a}^T \boldsymbol{\delta}}{\|\boldsymbol{\delta}\|_p}|} = \frac{|b|}{\max_{\boldsymbol{\delta}} |\frac{\boldsymbol{a}^T \boldsymbol{\delta}}{\|\boldsymbol{\delta}\|_p}|} = \frac{|b|}{\|\boldsymbol{a}\|_*} \tag{5}$$

It is well known from Hölder's inequality that $\|.\|_* = \|.\|_q$, where $q = \frac{p}{p-1}$.

## 2 SVM as a Special Case

In the special case of a linear classifier, our large margin formulation coincides with an SVM. Consider a binary classification task, so that $f_i(\boldsymbol{x}) \triangleq \boldsymbol{w}_i^T \boldsymbol{x} + b_i$, for $i = 1, 2$. Let $g(\boldsymbol{x}) \triangleq f_1(\boldsymbol{x}) - f_2(\boldsymbol{x}) = \boldsymbol{w}^T \boldsymbol{x} + b$ where $\boldsymbol{w} \triangleq \boldsymbol{w}_1 - \boldsymbol{w}_2$, $b \triangleq b_1 - b_2$. Recall from Eq. (7) (in the paper) that the distance to the decision boundary in our formulation is defined as:

$$\tilde{d}_{f,\boldsymbol{x},\{i,j\}} = \frac{|g(\boldsymbol{x})|}{\|\nabla_{\boldsymbol{x}} g(\boldsymbol{x})\|_2} = \frac{|\boldsymbol{w}^T \boldsymbol{x} + b|}{\|\boldsymbol{w}\|_2} , \tag{6}$$

In this (linear) case, there is a scaling redundancy; if the model parameter $(\boldsymbol{w}, b)$ yields distance $\tilde{d}$, so does $(c\boldsymbol{w}, cb)$ for any $c > 0$. For SVMs, we make the problem well-posed by assuming $|\boldsymbol{w}^T\boldsymbol{x} + b| \geq 1$ (the subset of training points that attains the equality are called support vectors). Thus, denoting the evaluation of $\tilde{d}$ at $\boldsymbol{x} = \boldsymbol{x}_k$ by $\tilde{d}_k$, the inequality constraint implies that $\tilde{d}_k \geq \frac{1}{\|\boldsymbol{w}\|_2}$ for any training point $\boldsymbol{x}_k$. The margin is defined as the smallest such distance $\gamma \triangleq \min_k \tilde{d}_k = \frac{1}{\|\boldsymbol{w}\|_2}$. Obviously, maximizing $\gamma$ is equivalent to minimizing $\|\boldsymbol{w}\|_2$; the well-known result for SVMs.

# 3 MNIST - Additional Results

Figure 1: Performance of MNIST models on noisy label tasks. In this plot and all others, "Xent" refers to cross-entropy.

Figure 2: Performance of MNIST models on generalization tasks. Gal etal. refers to the method of Gal et al. (2017).

Figure 3: Performance of MNIST models on white-box and black-box FGSM attacks. For the black-box, attacker is a cross-entropy trained model (best seen in PDF).

Figure 4: Performance of selected MNIST models on input Gaussian Noise with varying standard deviations.

Figure 5: Performance of CIFAR-10 models on FGSM adversarial examples (best seen in PDF).

## 4 FGSM/IFGSM Adversarial Example Generation

Given an input image $\boldsymbol{x}$, a loss function $L(x)$, the perturbed FGSM image $\hat{\boldsymbol{x}}$ is generated as $\hat{\boldsymbol{x}} \triangleq \boldsymbol{x} + \epsilon\ \mathrm{sign}(\nabla_{\boldsymbol{x}} L(\boldsymbol{x}))$. For IFGSM, this process is slightly modified as $\hat{\boldsymbol{x}}^k \triangleq \mathrm{Clip}_{\boldsymbol{x},\epsilon}(\hat{\boldsymbol{x}}^{k-1} + \alpha\,\mathrm{sign}(\nabla_{\boldsymbol{x}} L(\hat{\boldsymbol{x}}^k)))$, $\hat{\boldsymbol{x}}^0 = \boldsymbol{x}$, where $\mathrm{Clip}_{\boldsymbol{x},\epsilon}(\boldsymbol{y})$ clips the values of each pixel $i$ of $\boldsymbol{y}$ to the range $(\boldsymbol{x}_i - \epsilon, \boldsymbol{x}_i + \epsilon)$. This process is repeated for a number of iterations $k \geq \epsilon/\alpha$. The value of $\epsilon$ is usually set to a small number such as $0.1$ to generate imperceptible perturbations which can nevertheless cause the accuracy of a network to be significantly degraded.

## 5 CIFAR-10 - Additional Results

Figure 5 shows the performance of the CIFAR-10 models against FGSM black-box and white-box attacks.

## 6 MNIST Model Architecture and Hyperparameter Details

We use a $4$ hidden layer network. The first two hidden layers are convolutional, with filter sizes of $5 \times 5$ and $32$ and $64$ units. The hidden layers are of size $512$ each. We use a learning rate of $0.01$. Dropout is set to either $0$ or $0.2$ (hyperparameter sweep for each run), weight decay to $0$ or $0.005$, and $\gamma_l$ to $200$ or $1000$ (for margin only). The same value of $\gamma_l$ is used at all layers where margin is applied for ease of hyperparameter tuning. The aggregation operator (see (4) in the paper) is set to $\max$. For margin training, we add cross-entropy loss with a small weight of $1.0$. For example, as noise levels increase, we find that dropout hurts performance. The best performing validation accuracy among the parameter sweeps is used for reporting final test accuracy. We run for $50000$ steps of training with mini-batch size of $128$. We use either SGD with momentum or RMSProp. We fix $\epsilon$ at $10^{-6}$ for all experiments (see Equation 16 in the paper) - this applies to all datasets.

# 7 CIFAR-10 Model Architecture and Hyperparameter Details

We use the depth $58$, k=10 architecture from Zagoruyko & Komodakis (2016). This consists of a first convolutional layer with filters of size $3 \times 3$ and $16$ units; followed by 3 sets of residual units (9 residual units each). No dropout is used. Learning rate is set to $0.001$ for margin and $0.01$ for cross-entropy and hinge. with decay of $0.9$ every 2000 or 20000 steps (we choose from a hyperparameter sweep). $\gamma_l$ is set to either 5000, 10000 or 20000 (as for MNIST, the same value of $\gamma_l$ is used at all layers where margin is applied). The aggregation operator for margin is set to $\sum$. We use either SGD with momentum or RMSProp. For margin training, we add cross-entropy loss with a small weight of $1.0$.

# 8 Imagenet Model Architecture and Hyperparameter Details

We follow the architecture in Szegedy et al. (2016). We always use RMSProp for optimization. For margin training, we add cross-entropy loss with a small weight of $0.1$ and an additional auxiliary loss (as suggested in the paper) in the middle of the network.

# 9 Evolution of distance metric

Below we show plots of our distance approximation (see (7)) averaged over each mini-batch for 50000 steps of training CIFAR-10 model with $100\%$ of the data. This is a screengrab from a Tensorflow run (Tensorboard). The orange curve represents training distance, red curve is validation and blur curve is test. We show plots for different layers, for both cross-entropy and margin. It is seen that for each layer, (input, layer 7 and output), the margin achieves a higher mean distance to boundary than cross-entropy (for input, margin achieves mean distance about 100 for test set, whereas cross-entropy achieves about 70.). Note that for cross-entropy we are only *measuring* this distance.

Figure 6: Cross-entropy (top) Margin (bottom) mean distance to boundary for input layer of CIFAR-10.

Figure 7: Cross-entropy (top) Margin (bottom) mean distance to boundary for hidden layer 7 of CIFAR-10.

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

Figure 8: Cross-entropy (top) Margin (bottom) mean distance to boundary for output layer of CIFAR-10.