[Reviews · NeurIPS 2018]

Reviewer 1



The paper provided a novel and general definition of classifier margin, with the standard svm definition of margin as a special case. Analytical expression of the margin was derived, together with practical implementations to reduce computation. The experimental results also show the effectiveness of this large margin training criterion compared with standard cross entropy criterion. Overall, this paper is clearly written with strength both in theory and practice. Minor issue: In equation (8), (9) and (10), all x should be x_k.

Reviewer 2



This paper proposes a new method for encouraging large margin in the classification problems. In particular, this work is a generalization of SVM to deep models. The idea is interesting, and the method has been tested on many datasets and tasks. This paper has a few problems. 1. The paper should elaborate the optimization of Eqn 9 and 10, at least in the appendix. The optimization of such problems appears to be nontrivial. The short description in Sec 4.1 is not very easy to follow. Also, what is the time complexity of a forward and backward propagation? Is it much more expensive than optimization the cross-entropy loss? 2. Although I am not sure, I think the proposed large-margin method is very similar to adversarial training. The discussion of the similarity and difference may be interesting. 3. In the experiments, is the large-margin formulation (Eqn 10) applied to every layer or just a few selected layers? Also, Eqn 10 has a parameter $\gamma_l$. If the Eqn 10 is applied to many layers, will the hyper-parameter tuning be expensive? 4. As mentioned by Sec 4.1, the proposed method is expensive if there are many classes. The proposed method does not actually apply to the full ImageNet dataset. I am not sure whether the heuristic of using only one class $i \neq y_k$ is a reasonable approach. 5. Line 160 "... computation of second-order derivatives...": Do the 2nd-order derivatives even exist?

Reviewer 3



The paper proposes a loss function to help increase the margin of neural networks, which improves generalization and robustness of the model, and empirically shows better performance on MNIST, CIFAR-10 and ImageNet. The paper is well written, and the experiments are sufficient. The model outperforms classical loss such as cross entropy loss, which shows the effect of their loss function. On the other hand, CW attack is not tested in the Adversarial Perturbation experiments, which is more difficult to defend. Similar idea has been implemented in the previous paper "Formal Guarantees on the Robustness of a Classifier against Adversarial Manipulation", which weaken the novelty of this paper.